:ᴏ: PLOS | ONE

# Multimorbidity and complex multimorbidity in Brazilian rural workers

**Glenda Blaser Petarli[1], Monica Cattafesta[1], Monike Moreto Sant'Anna[2], Olívia Maria de Paula Alves Bezerra[3], Eliana Zandonade[1], Luciane Bresciani Salaroli** [ID][4]*

**1** Postgraduate Program in Collective Health, Health Sciences Center, Federal University of Espírito Santo, Vitória, Espírito Santo, Brazil, **2** Center for Health Sciences, Federal University of Espírito Santo, Vitória, Brazil, **3** Department of Family Medicine, Mental and Collective Health, Medical school, Federal University of Ouro Preto, Ouro Preto, Minas Gerais, Brazil, **4** Postgraduate program in Nutrition and Health, and Graduate Program in Collective Health, Center for Health Sciences, Federal University of Espírito Santo, Vitória, Espírito Santo, Brazil

* lucianebresciani@gmail.com

## Abstract

### Objective

To estimate the prevalence of multimorbidity and complex multimorbidity in rural workers and their association with sociodemographic characteristics, occupational contact with pesticides, lifestyle and clinical condition.

### Methods

This is a cross-sectional epidemiological study with 806 farmers from the main agricultural municipality of the state of Espírito Santo/Brazil, conducted from December 2016 to April 2017. Multimorbidity was defined as the presence of two or more chronic diseases in the same individual, while complex multimorbidity was classified as the occurrence of three or more chronic conditions affecting three or more body systems. Socio-demographic data, occupational contact with pesticides, lifestyle data and clinical condition data were collected through a structured questionnaire. Binary logistic regression was conducted to identify risk factors for multimorbidity.

### Results

The prevalence of multimorbidity among farmers was 41.5% (n = 328), and complex multimorbidity was 16.7% (n = 132). More than 77% of farmers had at least one chronic illness. Hypertension, dyslipidemia and depression were the most prevalent morbidities. Being 40 years or older (OR 3.33, 95% CI 2.06–5.39), previous medical diagnosis of pesticide poisoning (OR 1.89, 95% CI 1.03–3.44), high waist circumference (OR 2.82, CI 95% 1.98–4.02) and worse health self-assessment (OR 2.10, 95% CI 1.52–2.91) significantly increased the chances of multimorbidity. The same associations were found for the diagnosis of complex multimorbidity.

**Data Availability Statement:** All relevant data are within the manuscript and its Supporting Information files.

**Funding:** Yes. Financial support: Foundation for Research Support of Espírito Santo (FAPES) - Edict FAPES/ CNPq/ Decit-SCTIE-MS / SESA - PPSUS - n ˚ 05/2015. The funders had no role in study design, data collection and analysis, decision to publish, or preparation of the manuscript.

**Competing interests:** The authors have declared that no competing interests exist.

## Conclusion

We identified a high prevalence of multimorbidity and complex multimorbidity among the evaluated farmers. These results were associated with increased age, abdominal fat, pesticide poisoning, and poor or fair health self-assessment. Public policies are necessary to prevent, control and treat this condition in this population.

## Introduction

Exposure to dust, toxic chemicals, ultraviolet radiation, noise, and venomous animals in the daily routine of rural work represents potential sources of health problems for farmers [1]. Besides these, the transformations brought about by the mechanization and modernization of agricultural activities have modified the form of work organization in the field, with direct consequences to the physical and psychological domains, on the lifestyle and food consumption of these workers [2, 3].

This reality, aggravated by the reduced supply of health diagnosis and treatment services in rural areas [4], may increase farmers' vulnerability to chronic morbidity. Some evidence suggests worse health conditions and more disease among rural populations compared to other population groups [5,6,7]. It is noteworthy that these diseases may be isolated or coexist in the same individual, a condition known as multimorbidity [8]. Multimorbidity leads to a reduction in quality of life, higher mortality, polypharmacy, and an increase in the need for medical care, thus affecting health costs, and the productivity and functional capacity of individuals [9].

Knowing the distribution of diseases and the prevalence of multimorbidity in specific communities and populations is of fundamental importance for the planning and organization of health services and policies [10]. In this sense, when compared to the traditional criterion of classification of multimorbidity, the use of the concept of complex multimorbidity has been considered by some authors as a more effective way to identify people with priority care and plan the investment of health resources [11]. Nevertheless, no Brazilian study has been identified that has used this approach for the study of multimorbidity.

Given the above, and considering all the risk factors in the reality of rural work, the impact of chronic diseases on health, productivity and care costs, as well as the scarcity of data on multimorbidity in these professionals, this study aims to estimate the prevalence of multimorbidity and complex multimorbidity in rural workers and their association with sociodemographic characteristics, occupational contact with pesticides, lifestyle, and clinical condition.

## Materials and methods

### Data source

This is an cross-sectional epidemiological study derived from a larger study conducted in the municipality of Santa Maria de Jetibá, located in the state of Espírito Santo, southeastern Brazil, titled "Health condition and associated factors: a study of farmers in Espírito Santo—Agro-SaúdES", funded by the Espírito Santo Research Support Foundation (FAPES)—FAPES Notice / CNPq / Decit-SCTIE-MS / SESA—PPSUS—No. 05/2015.

### Study population

The original study involved a representative sample of male and female farmers who met the following inclusion criteria: aged 18 to 59 years, not pregnant, having agriculture as their main source of income, and being in full employment for at least six months.

## Sample size calculation

To identify the eligible farmers in the original study, data available in the individual and family records, as collected by the Family Health Strategy teams, were used to cover 100% of the 11 health regions in the municipality. Through this survey, we identified 7,287 farmers out of a total of 4,018 families. From this universe, the sample size calculation for the original project was performed considering 50% prevalence of outcomes (to maximize sample), 3.5% sampling error, and 95% significance level, making up a minimum sample of 708 farmers. 806 farmers were invited to compensate for possible losses. All sample size calculations were performed using the EPIDAT program (version 3.1). The participants were selected by a stratified lot, considering the number of families by health region and by Community Health Agent (CHA), in order to respect proportionality among the 11 regions and among the 80 CHAs. Only one individual per family was admitted, thus avoiding the interdependence of information. In case of refusal or non-attendance, a new participant was called from the reserve list, respecting the sex and the health unit of origin of the person who gave up/refused.

It should be noted that, due to the characteristics of the investigated municipality in which family farming predominates, the farmers who participated in this study had farming practices characterized by the predominance of polyculture and low degree of mechanization.

For the analytical developments proposed in this paper, the minimum sample size was calculated considering an estimated prevalence of multimorbidity in rural populations of 18.6% [12], 3% error, and a 95% confidence interval, resulting in a minimum required sample of 594 individuals. However, to improve sample representativeness and statistical relevance, we used data from all farmers who participated in the original project.

## Data collection

Data collection of the original study took place between December 2016 and April 2017 in the dependencies of the health units of the municipality. A semi-structured questionnaire was applied, containing questions about socioeconomic, demographic, and occupational characteristics, occupational contact with pesticides, lifestyle, eating habits, and health condition, including the presence of chronic diseases and self-rated health. All this information was obtained through self-report. Anthropometric measurements were also collected, such as waist circumference, hemodynamic data such as systolic blood pressure (SBP), diastolic blood pressure (DBP), and blood drawn for biochemical examinations for markers such as thyroid stimulating hormone (TSH) and total cholesterol and fractions. To obtain biochemical data, 10 mL of blood was collected by venipuncture after 12 hours of fasting.

Only the variables of interest for this article were selected.

## Variables selected for this study

Multimorbidity was evaluated in two different ways: through the traditional concept defined as the presence of two or more chronic diseases in the same individual (Multimorbidity $\geq 2$ CD) [8] and through the concept of "complex multimorbidity", classified as the occurrence of three or more chronic conditions affecting three or more body systems or different domains [13].

Chronic diseases were identified by counting morbidities reported by farmers from the question: "Has a doctor or other health professional ever told you that you had any of these diseases?". Chronic diseases investigated in this study were: arrhythmia, infarction, stroke, diabetes mellitus, herniated disk, arthrosis, Repetitive Strain Injuries/Work Related Musculoskeletal Disorders (RSI/WMSD), renal disease, Parkinson's, Alzheimer's, cirrhosis, infertility, cancer, thyroid diseases, asthma, bronchitis, and pulmonary emphysema. In addition to the

diseases referred to through self-report, we also considered the diagnoses of hypertension, dyslipidemia, thyroid disorders, and depression, performed through this research.

To determine the organic systems or domains affected according to each disease, we used the International Classification of Diseases– 11th revision (ICD-11), namely: circulatory system (hypertension, stroke, infarct, cardiac arrhythmia), endocrine, nutritional or metabolic disorders (diabetes, dyslipidemia, thyroid changes), musculoskeletal or connective tissue system (RSI/WMSD, arrhythmia), mental, behavioral or neurodevelopmental disorders (Alzheimer's, depression), genitourinary system (infertility, kidney diseases), digestive system (liver cirrhosis), pulmonary system (bronchitis, asthma, pulmonary emphysema), and neoplasms (cancer).

The classification of blood pressure levels was performed based on the values of SBP and DBP according to the classification established in the VII Brazilian Hypertension Guidelines [14]. Thus, subjects with SBP $\geq$ 140 mmHg and/or DBP $\geq$ 90 mmHg or who reported the use of blood pressure medications were considered hypertensive. These measurements were measured during the interview at least three times for each individual using the Omron® Automatic Pressure Monitor HME-7200, calibrated and validated by the National Institute of Metrology, Quality and Technology (INMETRO). To avoid interference with the results, subjects were instructed to sit and rest for about five minutes, empty their bladder and not consume food, alcohol, coffee or cigarettes for 30 minutes prior to the assessment. For data analysis, the average of two measurements was considered and a third measurement was performed whenever the difference between the first two was greater than 4 mmHg [15].

To investigate dyslipidemia, the levels of total cholesterol, HDL-c, LDL-c and triglycerides were measured. Total cholesterol and HDL cholesterol were determined, respectively, by the enzymatic colorimetric method with the Cholesterol Liquicolor Kit (In Vitro Diagnostica Ltda) and the Cholesterol HDL Precipitation Kit (In Vitro Diagnostica Ltda). To determine LDL cholesterol, we used the Friedewald formula [16]. Triglycerides were determined by the enzymatic colorimetric method with the Triglycerides Liquicolor mono® Kit (In Vitro Diagnostica Ltda). The results were classified according to the V Brazilian Guidelines on Dyslipidemias and Prevention of Atherosclerosis [16]. Individuals who reported the use of lipid-lowering drugs were also considered dyslipidemic.

In addition to self-report, the thyroid alteration was also evaluated by measuring the TSH through the chemiluminescence method. Farmers who had TSH values of 0.34 to 5.60 µUI/mL were considered as having "no thyroid alteration", and individuals that had values above or below the reference range were classified as "with thyroid alteration".

To evaluate the symptoms of depression, the Major Depressive Episode Module of the Mini-International Neuropsychiatric Interview (MINI) version 5.0 [17] was used. We considered "With Depression" farmers classified through the MINI with "Current Depression Episode" or "Recurrent Depression Episode".

Independent variables included socioeconomic variables (sex, age, race/color, marital status, schooling, socioeconomic class, and land tenure), occupational characteristics related to exposure to pesticides (use of Personal Protective Equipment, frequency and number of pesticides used), lifestyle (smoking, physical activity, alcohol consumption) and clinical conditions (previous intoxication by agrochemicals, waist circumference, and self-assessment of health). All these variables were collected by self-report.

Socioeconomic class was determined according to the Brazilian Economic Classification Criterion [18], in which A and B are the highest economic levels, C is intermediate, and D or E are low economic levels. Schooling was assessed by the number of years of study reported by the farmer.

Regarding lifestyle-related variables, all were obtained by self-report. It was considered that a "smoker" would be a farmer who reported smoking, an "ex-smoker" one who did not smoke, but who had smoked in the past, and a "non-smoker" would be a farmer who had reported never having smoked. Alcohol intake was assessed by asking, "How often do you drink alcohol?" Farmers who reported consuming alcohol, regardless of time or amount, were categorized as "Consuming." Those who reported not drinking alcohol were classified as "Not consuming". Farmers were also asked if they performed any other physical activities than those related to agricultural work. Answers were categorized as "Yes" or "No", regardless of the type, time, or intensity of the exercise performed.

Health self-assessment was assessed by the question "In general, compared to people your age, how do you consider your own health status?", assuming "very good", "good", "fair" and "poor." Subsequently, we categorized the variable as "good/very good" and "fair/poor". Waist circumference was classified according to the World Health Organization [19], considering values $\leq 94$cm for men and $\leq 80$cm for women as "without metabolic risk", and "increased metabolic risk" for the other values. To collect this measurement, a 1cm wide Sanny® brand inextensible tape measure was used in triple measurement. The subject was instructed to stand, arms outstretched and feet together. The tape was positioned at the smallest curvature located between the last costal arch and the iliac crest. When it was impossible to locate the smallest curvature, we used the midpoint between these two anatomical points as the reference.

### Statistical analyses

The absolute and relative frequencies of the independent variables were calculated according to the presence or absence of multimorbidity ($\geq 2$ CD) and multimorbidity complex outcomes. Then, the chi-square test was performed to verify the association between them. Variables with p-value $< 5\%$ in this test were included in the logistic regression analysis. The odds ratio was adjusted with respective 95% confidence intervals. The quality of the model was accounted for by the Hosmer-Lemeshow test.

The study was approved by the Research Ethics Committee of the Health Sciences Center of the Federal University of Espírito Santo, Opinion no. 2091172 (CAAE 52839116.3.0000.5060). All participants signed the Informed Consent Form.

## Results

Of the 806 participants, 790 individuals completed the study. Of these, 612 (77.4%) had at least one chronic disease (Fig 1). Hypertension, dyslipidemia and depression were the most prevalent conditions, affecting 35.8% (n = 283), 34.4% (n = 272) and 16.9% (n = 134), respectively, of the farmers. Pulmonary emphysema, hepatic cirrhosis, infertility, Parkinson's, stroke, infarction, and Alzheimer's were reported by less than 1% of the sample. When the affected systems were evaluated, we found that 42.7% (n = 338) of the changes referred to endocrine, nutritional or metabolic diseases, followed by the circulatory system (37.4%, n = 296) and mental, behavioral or neurodevelopmental disorders (16.9%, n = 134).

Multimorbidity ($\geq 2$ CD) was found in 328 farmers (41.5%), and complex multimorbidity in 132 (16.7%) of the sample.

In the bivariate analyses (Table 1), the sociodemographic variables associated to both multimorbidity ($\geq 2$ DC) and complex multimorbidity were the age group and socioeconomic class. Sex (p = 0.005), marital status (p = 0.012) and schooling (p = 0.001) were only associated with multimorbidity ($\geq 2$ CD).

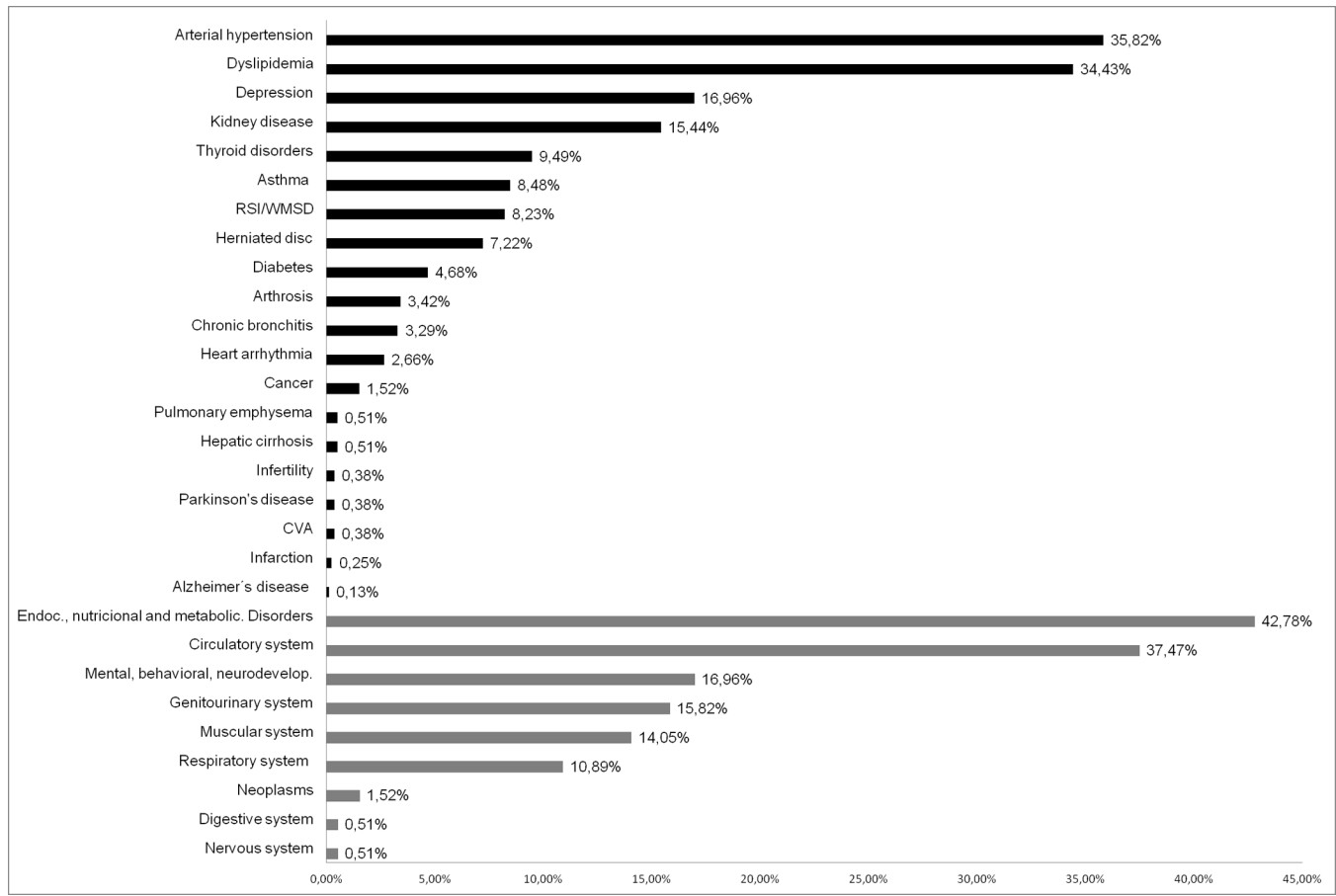

**Fig 1. Prevalence of chronic conditions expressed alone and according to organic system/ICD-11 domain affected in rural workers from Espírito Santo, Brazil.**

With regard to the occupational characteristics related to the use of pesticides, lifestyle and clinical condition, we verified that alcohol consumption, medical diagnosis of pesticide intoxication, waist circumference, and health self-assessment were associated with both outcomes (Table 2). Smoking was only associated with multimorbidity ($\geq$ 2 CD).

After a logistic regression analysis (Table 3), it was verified that being 40 years of age or older (OR 3.33, 95% CI 2.06–5.39), previous medical diagnosis of pesticide poisoning (OR 1.89, 95% CI 1.03–3.44), high waist circumference (OR 2.82, 95% CI 1.98–4.02), and fair or poor health self-assessment (OR 2.10, 95% CI 1.52–2.91) significantly increased the chances of multimorbidity ($\geq$ 2 DC).

The same associations were found for the diagnosis of complex multimorbidity.

## Discussion

This is the first Brazilian study to estimate the prevalence of multimorbidity in rural workers and to use the complex multimorbidity criterion to determine this outcome. The representative sample, stratified and randomly selected, allows us to extrapolate the results to the target population.

Agriculture is often described as an occupation that promotes health, being associated with the image of a healthy lifestyle with exposure to nature, outdoors, physical effort, and a diet

**Table 1. Prevalence of multimorbidity ($\geq$ 2 CD) and complex multimorbidity according to sociodemographic characteristics of farmers from Espírito Santo, Brazil.**

| Variable | Sample | Multimorbidity ($\geq$ 2 CD) | | | Complex Multimorbidity | | |
|---|---|---|---|---|---|---|---|
| | n (%) | % | IC 95%[a] | p-value[b] | % | IC 95%[a] | p-value[b] |
| **Sex** | | | | | | | |
| Male | 413 (52.3) | 36.8 | (33.4–40.2) | **0.005[c]** | 15.5 | (13.0–18.0) | 0.339 |
| Female | 377 (47.7) | 46.7 | (43.0–50.0) | | 18.0 | (15.3–20.7) | |
| **Age Group** | | | | | | | |
| Up to 29 years | 213 (27.0) | 23.5 | (20.5–26.5) | **0.000[c]** | 9.4 | (7.4–11.4) | **0.000[c]** |
| 30 to 39 years | 231 (29.2) | 33.8 | (30.5–37.1) | | 12.1 | (9.8–14.4) | |
| 40 or more | 346 (43.8) | 57.8 | (54.4–61.2) | | 24.3 | (21.3–27.3) | |
| **Race / Color** | | | | | | | |
| White | 702 (88.9) | 41.2 | (37.8–44.6) | 0.572 | 16.2 | (13.6–18.8) | 0.318 |
| Non-White | 88 (11.1) | 44.3 | (40.8–47.8) | | 20.5 | (17.7–23.3) | |
| **Marital status** | | | | | | | |
| Not married | 59 (7.5) | 27.1 | (24.0–30.2) | **0.012[c]** | 13.6 | (11.2–16.0) | 0.246 |
| Married/Living with partner | 678 (85.8) | 41.7 | (38.3–45.0) | | 16.4 | (13.8–19.0) | |
| Separated/Divorced/Widowed | 53 (6.7) | 54.7 | (51.2–58.2) | | 24.5 | (21.5–27.5) | |
| **Schooling** | | | | | | | |
| Less than 4 years | 533 (67.5) | 46.2 | (42.7–49.7) | **0.001[c]** | 18.0 | (15.3–20.7) | 0.367 |
| 4 to 8 years | 173 (21.9) | 33.5 | (30.2–36.8) | | 13.9 | (11.5–16.3) | |
| More than 8 years | 84 (10.6) | 28.6 | (25.4–31.8) | | 14.3 | (11.9–16.7) | |
| **Socioeconomic class** | | | | | | | |
| Class A or B | 58 (7.3) | 31.0 | (27.8–34.2) | **0.033[c]** | 6.9 | (5.1–8.7) | **0.050[c]** |
| Class C | 395 (50.0) | 39.0 | (35.6–42.4) | | 15.9 | (13.4–18.4) | |
| Class D or E | 337 (42.7) | 46.3 | (42.8–49.8) | | 19.3 | (16.5–22.1) | |
| **Land ownership** | | | | | | | |
| Owner | 609 (77.1) | 41.4 | (38.0–44.8) | 0.884 | 15.6 | (13.1–18.1) | 0.125 |
| Non-Owner | 181 (22.9) | 42.0 | (38.6–45.4) | | 20.4 | (17.6–23.2) | |

a Confidence Interval.

b Chi-square test.

[c] Statistically significant value (p <0.05).

based on natural foods [3]. However, the results reflect a different reality. Eight out of 10 farmers had at least one chronic disease, more than 40% had two or more, and around 17% had three or more, affecting at least three or more organic systems or ICD-11 domains.

Among the chronic conditions analyzed, there was a predominance of arterial hypertension and dyslipidemia, similar to other multimorbidity studies performed in Brazil [20] and in countries such as Portugal [21] and Australia [22]. These two morbidities were also more frequent in disease pattern studies conducted for the population of the United States [23] and New York State [24]. In a systematic review involving studies from 16 European countries [9] hypertension also occupied a prominent position, as well as countries such as China, Finland, Ghana, Russia, South Africa [25] and in four Greater Mekong countries [26]. The prevalence of these diseases is also observed when evaluating multimorbidity studies with the elderly [27, 28]. These values, however, are above the estimate for the Brazilian population through wide-ranging studies such as VIGITEL (24.1%) [29] and National Household Sample Survey—PNAD (20.9%) [30].

**Table 2. Prevalence of multimorbidity (≥ 2 CD) and complex multimorbidity according to occupational characteristics related to the use of pesticides, lifestyle and clinical condition of farmers from Espírito Santo, Brazil.**

| Variables | Sample | Multimorbidity (≥ 2 CD) | | | Complex Multimorbidity | | |
|---|---|---|---|---|---|---|---|
| | n | % | IC 95%[a] | p-value[b] | % | IC 95%[a] | p-value[b] |
| **Type of occupational contact with pesticide** | | | | | | | |
| Direct | 550 (69.6) | 40.7 | (37.3–44.1) | 0.494 | 15.3 | (12.8–17.8) | 0.101 |
| Indirect/Non-Contact | 240 (30.4) | 43.3 | (39.8–46.8) | | 20.0 | (17.2–22.8) | |
| **Total number of pesticides used** | | | | | | | |
| None | 240 (32.0) | 43.3 | (39.8–46.8) | 0.502 | 20.0 | (17.1–22.9) | 0.268 |
| 1 to 5 types of pesticides | 223 (29.7) | 42.6 | (39.1–46.1) | | 15.7 | (13.1–18.3) | |
| More than 5 pesticides | 287 (38.3) | 38.7 | (35.2–42.2) | | 15.0 | (12.4–17.6) | |
| **Use of PPE** | | | | | | | |
| Do not use PPE/Incomplete PPE | 380 (49.2) | 42.6 | (39.1–46.1) | 0.194 | 16.6 | (14.0–19.2) | 0.073 |
| Complete PPE | 152 (19.7) | 34.9 | (31.5–38.3) | | 11.2 | (11.2–13.4) | |
| Without direct contact | 240 (31.1) | 43.3 | (39.8–46.8) | | 20.0 | (17.2–22.8) | |
| **Frequency of contact with pesticide** | | | | | | | |
| Daily/Weekly | 453 (61.4) | 40.8 | (37.3–44.3) | 0.717 | 15.2 | (12.6–17.8) | 0.235 |
| Monthly/Yearly | 206 (27.9) | 43.7 | (40.1–47.3) | | 17.5 | (14.8–20.2) | |
| Without contact | 79 (10.7) | 44.3 | (40.7–47.9) | | 22.8 | (19.8–25.8) | |
| **Smoking** | | | | | | | |
| Non-smoker | 665 (84.2) | 39.8 | (36.4–43.2) | 0.028[c] | 15.9 | (13.4–18.4) | 0.181 |
| Smoker or ex-smoker | 125 (15.8) | 50.4 | (46.9–53.9) | | 20.8 | (18.0–23.6) | |
| **Practices physical activity** | | | | | | | |
| No | 669 (84.7) | 42.9 | (39.4–46.4) | 0.064 | 17.2 | (14.6–19.8) | 0.394 |
| Yes | 121 (15.3) | 33.9 | (30.6–37.2) | | 14.0 | (11.6–16.4) | |
| **Alcohol consumption** | | | | | | | |
| Does not consume | 444 (56.2) | 46.8 | (43.3–50.3) | 0.001[c] | 21.2 | (18.3–24.1) | 0.000[c] |
| Consumes | 346 (43.8) | 34.7 | (31.4–38.0) | | 11.0 | (8.0–13.2) | |
| **Medical diagnosis of poisoning by pesticides** | | | | | | | |
| Yes | 59 (7.5) | 57.6 | (54.1–61.1) | 0.010[c] | 32.2 | (28.9–35.5) | 0.001[c] |
| No | 729 (92.5) | 40.3 | (36.9–43.7) | | 15.5 | (13.0–18.0) | |
| **Waist circumference** | | | | | | | |
| Without metabolic risk | 384 (48.7) | 26.0 | (22.9–29.1) | 0.000[c] | 9.4 | (7.4–11.4) | 0.000[c] |
| Increased metabolic risk | 405 (51.3) | 56.3 | (52.8–59.8) | | 23.7 | (20.7–26.7) | |
| **Health self-assessment** | | | | | | | |
| Good/ Very good | 459 (58.1) | 32.2 | (28.9–35.5) | 0.000[c] | 10.5 | (8.4–12.6) | 0.000[c] |
| Fair/Poor | 331 (41.9) | 54.4 | (50.9–57.9) | | 25.4 | (22.4–28.4) | |

[a] Confidence Interval.

[b] Chi-square test.

[c] Statistically significant value (p <0.05).

By analyzing the presence of chronic diseases according to the organic system or affected area, it was found that the most frequent ones were endocrine, nutritional or metabolic diseases, due to the high rates of dyslipidemia, diabetes and thyroid disorders, and the circulatory system, due to arterial hypertension. In Spain, a research project with more than one million patients also identified a predominance of alterations in these two systems, especially in individuals over 45 years old [31]. In Ethiopia, however, musculoskeletal system diseases were the most prevalent, affecting about 20% of the sample [32]. In an Australian study, there were 32.4% alterations involving the circulatory system, 32.1% of musculoskeletal and connective

**Table 3.** Association between multimorbidity (≥ 2 DC), complex multimorbidity and socio-demographic characteristics, occupational contact with pesticides, lifestyle and clinical condition in farmers from Espírito Santo, Brazil.

| Variables | Multimorbidity (≥ 2 DC) | | | | Complex Multimorbidity | | | |
|---|---|---|---|---|---|---|---|---|
| | p-value[a] | OR adjusted[b] | LL[c] | UL[d] | p-value[a] | OR adjusted[b] | LL[c] | UL[d] |
| **Sex** | | | | | | | | |
| Male | | 1 | | | | | | |
| Female | 0.854 | 1.037 | 0.702 | 1.533 | | | | |
| **Age Group** | | | | | | | | |
| Up to 29 years | | 1 | | | | 1 | | |
| 30 to 39 years | 0.131 | 1.438 | 0.897 | 2.305 | 0.682 | 1.141 | .606 | 2.149 |
| 40 or more | **0.000[e]** | **3.336** | **2.065** | **5.390** | **0.004[e]** | **2.250** | **1.295** | **3.909** |
| **Marital status** | | | | | | | | |
| Not married | | 1 | | | | | | |
| Married / Living with partner | 0.999 | 1.000 | 0.511 | 1.957 | | | | |
| Separated / Divorced / Widowed | 0.951 | 1.028 | 0.418 | 2.527 | | | | |
| **Schooling** | | | | | | | | |
| More than 8 years | 0.643 | 1.162 | 0.617 | 2.188 | | | | |
| 4 to 8 years | 0.971 | 1.011 | 0.559 | 1.829 | | | | |
| Less than 4 years | | 1 | | | | | | |
| **Socioeconomic class** | | | | | | | | |
| Class A or B | | 1 | | | | 1 | | |
| Class C | 0.652 | 1.165 | 0.600 | 2.263 | 0.174 | 2.115 | 0.719 | 6.224 |
| Class D or E | 0.256 | 1.488 | 0.749 | 2.956 | 0.117 | 2.378 | 0.805 | 7.025 |
| **Smoking** | | | | | | | | |
| Non-smoker | | 1 | | | | | | |
| Smoker or ex-smoker | 0.070 | 1.534 | 0.965 | 2.438 | | | | |
| **Alcohol consumption** | | | | | | | | |
| Does not consume | | 1 | | | | 1 | | |
| Consumes | 0.314 | 0.835 | 0.588 | 1.186 | 0.069 | 0.666 | 0.429 | 1.032 |
| **Medical diagnosis of poisoning by pesticides** | | | | | | | | |
| No | | 1 | | | | 1 | | |
| Yes | **0.038[e]** | **1.891** | **1.037** | **3.449** | **0.005[e]** | **2.474** | **1.319** | **4.638** |
| **Waist Perimeter** | | | | | | | | |
| Without metabolic risk | | 1 | | | | 1 | | |
| Increased metabolic risk | **0.000[e]** | **2.829** | **1.986** | **4.029** | **0.001[e]** | **2.142** | **1.370** | **3.349** |
| **Health Self-Assessment** | | | | | | | | |
| Good/ Very good | | 1 | | | | 1 | | |
| Fair/Poor | **0.000[e]** | **2.107** | **1.524** | **2.913** | **0.000[e]** | **2.248** | **1.493** | **3.384** |

[a] Binary Logistic Regression. Enter Method.

[b] Odds Ratio.

[C] Lower Limit– 95% Confidence interval.

[d] Upper Limit– 95% Confidence interval.

[e] Statistically significant value (p <0.05).

Hosmer-Lemeshow = 0.795 (Multimorbidity ≥ 2 DC) and 0.701 (Complex Multimorbidity)

tissue, and 30.7% of endocrine, nutritional and metabolic alterations [33]. These results corroborate the three globally most common multimorbidity groups, composed of "metabolic disorders", including diabetes, obesity and hypertension, "mental-articular disorders",

including arthritis and depression, and "cardio-respiratory" including angina, asthma and chronic obstructive pulmonary disease [25].

The high number of farmers with chronic conditions involving mental, behavioral or neurodevelopmental disorders, especially due to the high prevalence of depression in these workers, is worth highlighting. Depressive disorders were also among the most frequent in the study by Prazeres and Santiago [21] and the study by Harrison et al. [33], in which mental/psychological changes were found in 26.7% of the sample.

The prevalence of multimorbidity presented by rural workers was higher than estimated for the Brazilian population through the World Health Survey (13.4%, 95% CI 12.4–14.5) [34] and the National Health Survey [12], in which the expected multimorbidity was 18.6% (95% CI 17.2–20.0%) in rural areas and 22.8% (95% CI 22–23.5%) in Brazilian urban areas. It was also higher than the prevalence found in developed countries, such as Portugal (38.3%) [35], Spain (20%) [36], Canada (12,9%) [37], Denmark (22%) [38], and Belgium (22.7%) [39], and, in middle-income countries, where 12.6% (Mexico), 19.4% (Russia), and 10.4% (South Africa) of the 40–49 year-old population reported two or more chronic diseases [40]. In a study involving six countries in South America and the Caribbean, the self-reported multimorbidity ranged from 12.4% in Colombia to 25.1% in Jamaica [41]. It is estimated that between 16% and 57% of adults in developed countries suffer from more than one chronic condition [42]. In European health systems, the estimated prevalence of multimorbidity was 33% in 2015 [43]. A systematic review by Nguyen et al. [44] involving only community studies found a combined global prevalence of multimorbidity of 33.1%. Among the 37 representative studies of developed countries involved in this review, the lowest prevalence of multimorbidity was identified in Hong Kong (3.5%) and the highest in Russia (70%). Among developing countries, the lowest percentage identified was in 26 Indian villages (1%) and the highest prevalence was in China (90%) [44]. We emphasize that the methodological differences, especially those related to the target population and the diagnostic criterion of multimorbidity, limit the comparison and interpretation of the results.

With respect to the complex multimorbidity, few studies are available in international literature using this methodology. An Australian study estimated that 25.7% of the population had two or more chronic diseases and 12.1% showed complex multimorbidity [33]. This methodology shows itself as a more discriminatory measure, and among farmers, reduced the prevalence of multimorbidity compared to the criterion of two or more diseases. Harrison et al. [11] argue that counting affected body systems instead of evaluating individual chronic conditions has the advantage of more carefully identifying patients who need more complex care, as well as the number and types of specialized health services that are necessary for such assistance, thus being a more useful and effective way of planning actions and investments in health [11].

The only sociodemographic factor that remained associated with multimorbidity, regardless of the form of evaluation of this outcome, was age. This association is well documented in the literature. In Canada, the prevalence of multiple diseases increased from 12.5% in the younger age group (18–24 years) to 63.8% in the more advanced ones ($\geq$ 65 years) [45]. The onset of chronic diseases with increasing age seems to be related to the physiological imbalance and general senescence in multiple organs that aging causes [46]. This influence can be seen in comparison with the significant increase in multimorbidity in studies conducted with the elderly. Nunes et al. [47], analyzing a representative national sample of the non-institutionalized population, identified a prevalence of 82.4% of multimorbid individuals (CI 95% 78.5–85.7%) among older adults aged 80 years or older. In Southern Brazil, the estimate was 93.4% in the city of Pelotas [48] and 81.3% in Bagé [49]. In Canada, the overall prevalence of multimorbidity in the older age group ($\geq$ 85 years) was 58.6% higher when compared to younger age groups [50]. A marked difference was also found in the study by Puth et al. [51] in

Germany, where the prevalence of this condition increased from 7% in individuals aged 18–29 to 77.5% in those aged 80 and older.

Although there is a large amount of evidence that the occurrence of multimorbidity is higher in females and at low socioeconomic and educational levels [52], the association with socioeconomic variables is very heterogeneous between studies [53]. As with our results, the lack of association with income [26], education and gender [5] has also been documented. Several factors may be related to these results. Among them, we can mention the homogeneity of the rural population investigated in relation to income (92.7% belonged to lower socioeconomic classes—C, D or E), education (89.4% had fewer than 8 years of schooling) and marital status (85.8% were married or living with a partner), compared with the urban population, which generally has more heterogeneous strata, and is therefore more differentiated. This homogeneity of the analyzed population may have compromised the identification of statistically significant differences between strata.

Regarding gender, considering that in rural areas there is limited access to health services [54], there may have been under-reporting in the diagnosis of chronic diseases, especially among women, who generally use health services more often than men. This may have led to a reduction in self-reported disease among women and consequently the absence of statistical association between genders. In addition, the frequency of some diseases is known to vary by gender [55]. In this sense, the methodological differences regarding the type and quantity of diseases to be considered in each study for classification of multimorbidity have a direct influence on the results found by each author [56]. As an example, we can mention the study by Pengpid and Peltzer [26] that identified a higher prevalence of multimorbidity in men due to the inclusion of smoking and alcoholism among the evaluated chronic conditions. The methodology used for disease identification may also have contributed to the difference between the results [56, 57]. In the study by Guerra et al. [58], for example, gender was not associated with multimorbidity measured from administrative data, but was associated with self-reported multimorbidity, regardless of the cutoff point adopted. For this reason, the fact that we used both self-reported data as well as biochemical and hemodynamic data may justify the differences in association found when compared to other studies, which mostly use self-reported data.

In addition to methodological differences, the lack of association with some sociodemographic variables may be due to the presence of factors that contribute more closely to the development of multimorbidity, such as waist circumference, previous pesticide poisoning or other factors not intrinsic to agricultural activity within the scope of this study. Further studies involving farmers are needed to better understand the risk factors present in daily agricultural work, thus facilitating the comparison of results.

This study, however, strengthens the evidence of the association between the accumulation of visceral fat and the occurrence of chronic diseases. In addition to reflecting the level of central adiposity, high waist circumference is also directly related to excess body fat, and is considered a major risk factor for the early development of various morbidities, including hypertension, diabetes, dyslipidemias, and cancers [59]. Corroborating these results, a cohort conducted in the United Kingdom concluded that overweight participants were 25% more likely to have at least one of 11 assessed health conditions compared to normal weight subjects. In obese patients, the odds increased to 54%, 81% and 124% for categories I, II and III, respectively [60]. Similarly, different disease patterns identified in the Brazilian population were also associated with obesity [54]. In low- and middle-income countries, the prevalence of multimorbidity increased 5.78 fold in obese individuals when compared to those of normal weight [61].

In addition to the negative influence of age and waist circumference on the occurrence of multiple diseases, previous poisoning by pesticides also seems to be related to this condition, increasing by 1.89 and 2.47 the chance of occurrence of multimorbidity and complex multimorbidity among workers in rural areas, respectively. We highlight that there are several harmful health effects that have been related to the use of pesticides, among them mental disorders, respiratory, and autoimmune diseases [62]. Farmers who have reported being poisoned with pesticides may be more chronically exposed to these products and, therefore, more likely to show the cumulative deleterious effects of this exposure. The comparison of this result becomes limited, since other similar studies in literature were not found. It should be emphasized that the association between the outcome and the variables of exposure, intensity and frequency of contact with pesticides may not have been evidenced, due to the limitations of cross-sectional studies, when compared to cohort studies, to evaluate the oscillations in occupational exposure years.

Another factor associated with the higher prevalence of multimorbidity was the health self-assessment. Fair or poor health perception doubled the chances of occurrence of multimorbidity (CD $\geq$ 2) (OR 2.10, 95% CI 1.52–2.91) or complex multimorbidity (OR 2.24, 95% CI 1.49–3.38) among farmers. In European countries, the increased number of chronic diseases was also associated with a higher probability of reporting poor/fair health self-perception (OR = 2.13, 95% CI 2.03–2.24) [9]. The same association was found in countries in South America and the Caribbean [41], in the rural population of South Africa [63], and in Myanmar [57].

Thus, we verified that the rural population analyzed showed alarming rates, not only of a single chronic condition, but of multiple conditions. The occurrence of multiple diseases has been associated with aging, being overweight, the way farmers perceive their health, and occupational exposure to agrochemicals. It is also worth noting that, despite the fact that it was not within the scope of this research, it is known that factors such as the difficulty of access to health services and specialized treatments, which are common in rural communities, further increase the vulnerability of these workers to the development of multimorbidity.

Considering the serious economic, social and health implications of the presence of multiple chronic diseases in people of working age [60], it is necessary to re-examine the focus of the health system, which currently does not seem well suited to the new medical and social reality that the multimorbidity presents. As strategies and public policies must ensure holistic care, implementing actions that consider the particularities and vulnerabilities of this community, as well as stimulating self-care, controlling modifiable risk factors and adopting healthy behaviors [32]. Also, the training of health teams to attend multimorbid patients is essential, as well as the elaboration of clinical protocols for multiple diseases, and, especially, effective allocation of financial resources [64]. In this sense, although there is a great value in measuring the occurrence of chronic conditions in an individualized way, complex multimorbidity, through the measurement of the patterns of the bodily systems affected by chronic conditions, seems to be a good tool to screen, select and align services and prioritize resources more effectively to patients with greater need [11].

Among the limitations of this study, we emphasize that diseases identified through self-report may be subject to the under-reporting of diagnosis or memory bias. Despite the extensive list of diseases included, some chronic conditions may not have been identified. The lack of standardization on the way to evaluate multimorbidity and the unavailability in the literature of articles on this theme involving farmers limited the comparison of the results. Because this is a cross-sectional study, reverse causality cannot be disregarded in the data interpretation. It should be noted that the prevalence of multimorbidity may be under-reported since patients with more severe conditions may not have participated in the study.

Despite these limitations, it is worth noting the unprecedented nature of the study in relation to the involved target population, the adoption of the complex multimorbidity criterion and the included variables, such as waist circumference and pesticide intoxication, in articles of this theme. We sought in this study to involve a representative sample to allow extrapolation of the results to farmers with similar profile. In addition, in order to minimize the errors of underdiagnosis, the identification of diseases occurred, both through self-report, as well as through laboratory and hemodynamic measures.

## Conclusions

We identified a high prevalence of multimorbidity and complex multimorbidity among the evaluated farmers. Factors associated with these outcomes in this population were increased age, high waist circumference, history of pesticide intoxication, and poor or fair health self-assessment. Considering the serious physical, functional, psychological and economic implications of multimorbidity, it is fundamentally important to plan economic, social and health policies aimed at controlling, monitoring and treating this condition in this professional category. In addition, new research is needed to evaluate in more detail the impacts that the risk factors identified in this study may have on the health of rural workers, especially those resulting from being overweight and from occupational exposure to agrochemicals, both of which are associated in this study with the presence of multiple diseases.

## Supporting information

**S1 Database.**
(XLSX)

**S1 Table. Database variable codes.**
(DOCX)

## Acknowledgments

To all partner institutions especially the Municipal Health Secretariat of Santa Maria de Jetibá and the farmers who participated in the study.

## Author Contributions

**Conceptualization:** Glenda Blaser Petarli, Monica Cattafesta, Monike Moreto Sant'Anna, Olívia Maria de Paula Alves Bezerra, Eliana Zandonade, Luciane Bresciani Salaroli.

**Data curation:** Glenda Blaser Petarli, Monica Cattafesta, Monike Moreto Sant'Anna.

**Formal analysis:** Glenda Blaser Petarli.

**Funding acquisition:** Luciane Bresciani Salaroli.

**Investigation:** Glenda Blaser Petarli, Monica Cattafesta, Olívia Maria de Paula Alves Bezerra.

**Methodology:** Glenda Blaser Petarli, Monica Cattafesta, Eliana Zandonade, Luciane Bresciani Salaroli.

**Project administration:** Glenda Blaser Petarli.

**Resources:** Glenda Blaser Petarli.

**Software:** Glenda Blaser Petarli, Eliana Zandonade.

**Supervision:** Glenda Blaser Petarli, Olívia Maria de Paula Alves Bezerra, Luciane Bresciani Salaroli.

**Validation:** Glenda Blaser Petarli, Eliana Zandonade.

**Visualization:** Glenda Blaser Petarli, Monica Cattafesta, Monike Moreto Sant'Anna, Olívia Maria de Paula Alves Bezerra, Eliana Zandonade.

**Writing – original draft:** Glenda Blaser Petarli, Monica Cattafesta, Monike Moreto Sant'Anna, Olívia Maria de Paula Alves Bezerra, Eliana Zandonade, Luciane Bresciani Salaroli.

**Writing – review & editing:** Glenda Blaser Petarli, Monica Cattafesta, Monike Moreto Sant'Anna, Olívia Maria de Paula Alves Bezerra, Eliana Zandonade, Luciane Bresciani Salaroli.

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
