## [Decision Letter · Decision Letter 0]

9 Sep 2019

PONE-D-19-17072

Multimorbidity and complex multimorbidity in Brazilian rural workers

PLOS ONE

Dear Dr. Salaroli,

Thank you for submitting your manuscript to PLOS ONE. After careful consideration, we feel that it has merit but does not fully meet PLOS ONE’s publication criteria as it currently stands. Therefore, we invite you to submit a revised version of the manuscript that addresses the points raised during the review process.

In particular, please address the point made by Reviewer #1 on the lack of socio-demographic correlates. I suspect the reviewer is expecting to see multimorbidity occurring mainly in those with lower socio-economic status, and also those who are older. Please comment on whether there are special circumstances for the cohort studied that mask these factors. Please also provide more details on the sampling strategy and statistical procedures, in case this might be the result of a simple mistake.

We would appreciate receiving your revised manuscript by Oct 24 2019 11:59PM. To enhance the reproducibility of your results, we recommend that if applicable you deposit your laboratory protocols in protocols.io, where a protocol can be assigned its own identifier (DOI) such that it can be cited independently in the future. For instructions see: http://journals.plos.org/plosone/s/submission-guidelines#loc-laboratory-protocols

We look forward to receiving your revised manuscript.

Kind regards,

Siew Ann Cheong, Ph.D.

Academic Editor

PLOS ONE

Journal Requirements:

1. Please include additional information regarding the semi-structured questionnaire used in the study and ensure that you have provided sufficient details that others could replicate the analyses. For instance, if you developed a questionnaire as part of this study and it is not under a copyright more restrictive than CC-BY, please include a copy, in both the original language and English, as Supporting Information.

2. Please provide further details regarding how participants were recruited.

Reviewers' comments:

Reviewer's Responses to Questions

**Comments to the Author**

1. Is the manuscript technically sound, and do the data support the conclusions?

Reviewer #1: Yes

Reviewer #2: Yes

2. Has the statistical analysis been performed appropriately and rigorously? 

Reviewer #1: Yes

Reviewer #2: Yes

3. Have the authors made all data underlying the findings in their manuscript fully available?

Reviewer #1: Yes

Reviewer #2: Yes

4. Is the manuscript presented in an intelligible fashion and written in standard English?

Reviewer #1: Yes

Reviewer #2: No

5. Review Comments to the Author

Reviewer #1: In the present manuscript Petarli and colleagues aimed to estimate the prevalence of multimorbidity and complex multimorbidity in rural workers and their association with sociodemomographic, occupational, and clinical correlates. The article is quite interesting as it provides estimates based on a population with quite specific lifestyle and occupational exposure to specific agents (e.g. pesticides), which might require targeted interventions/public health policies. What I found quite important to highlight is that in the multivariable model, in contrast to previous literature, almost no socio-demographic correlates (with the exception of age, which is a well-established strong predictor of multimorbidity) remained significant, while all clinical ones were found associated with the likelihood of having multimorbidity and I would suggest authors to elaborate more on this (and possible explanations).

Other points:

- Introduction and discussion are sufficiently well-written but to make comparisons even more relevant the cited literature should be enriched with even more recent published papers on the epidmiology of multimorbidity (especially in western countries)

- Further details on the sampling strategy should be provided. Authors provide a SS calculation to detect selected outcomes (which should also be expanded with additional information to allow reader to better understand and repeat calculations) but I doubt that this survey was originally based on this SS calculation, please explain and elaborate more on the sampling in general. Additionally, authors do no mention the use of survey weights: does this mean that non-respondent problem was encountered?

- The authors adopt an empirical approach for the inclusion of covariates in the multirvariable models which is somehow in contrast with the current tendency to select covariates mostly by literature research. Please justify this approach

Reviewer #2: General comments: Authors have highlighted a very pertinent issues on multimorbidity and its complexities. The overall article content is clear; however,it needs to be reviewed and copy-edited for English language to make language of the article clear, correct before re-submitting.

Specific comments:

1. Abstract conclusion is not clear and need rephrasing for better clarity.

2. line 100-105: The sentences are confusing and not clear. It would be better to rewrite the sentences with more clarity. It is not clear what authors want to convey by saying "lack of standardization regarding the most effective way to diagnose this condition."

3. The study is based on data from "Condition of health and associated factors: a study in farmers of Espírito Santo - AgroSaúdES". As a part of better clarity it would be appropriate to provide a brief description of the parent study: how the sampling was done, how data were collected and what information collected in the parent study.

4. Line 120-121: Need rephrasing!

5. Sample size calculation: Authors have mentioned that they included 806 sample for their study which would provide 96.9% power to detect the outcome. This is not clear. Authors should provide a detail description of the sample size calculation in a separate section or within the section of study population.

6. It would be better to provide a more detail information on how and what information were collected under Anthropometric, hemodynamic and biochemical data.

7. line 157: what is the full form of RSI/WMSD

8. line no.170- The classification of Blood pressure level and not the "pressure level "

9. Line 197: lifestyle (smoking, physical activity, alcohol consumption)- How these lifestyle factors were elicited and examined?

10. line 206-208: Authors have categorised waist circumference based on WHO classification and termed it as without cardiovascular risk and increased cardiovascular risk. However, this WHO classification relates to metabolic risk rather than CVD risk.

11. Is there any other anthropometric measurements such as height and weight (BMI) were collected in the survey and if collected why authors have not included in the study.

12. Line 368: What are the types of diseases when you mentioned the word "disease" in this sentence?

13. References:

a. Line 450: link not working

b. References should be checked for uniformity in the formating style and journal names.

6. PLOS authors have the option to publish the peer review history of their article (what does this mean?). If published, this will include your full peer review and any attached files.

Reviewer #1: Yes: Raffaele Palladino

Reviewer #2: No

---

## [Author Response · Author response to Decision Letter 0]

26 Sep 2019

Dear Editor,

We appreciate all the contributions made by the reviewers. We reviewed the entire manuscript and followed all suggestions. For ease of identification, corrections have been marked in red in the article and we have answered questions one by one in this letter.

Regarding the lack of association between sociodemographic factors and multimorbidity (except age), I inform you that, as requested, we insert possible explanations for these results. We emphasize, however, that although less frequent, there are similar results already documented in the literature [16] [165].

We also report that the article was reviewed by a research expert at Proof Reading Service (https://www.proof-reading-service.com/en/). The certificate provided by the company has been attached to the PLOS ONE platform.

Sincerely Regards,

The authors.

Journal Requirements:

ANSWER: We review and follow the PLOS ONE's style.

1. Please include additional information regarding the semi-structured questionnaire used in the study and ensure that you have provided sufficient details that others could replicate the analyses. For instance, if you and it is not under a copyright more restrictive than CC-BY, please include a copy, in both the original language and English, as Supporting Information.

ANSWER: Further details on the methodological processes were included. 

2. Please provide further details regarding how participants were recruited.

ANSWER: The methodology has been rewritten in more detail.

Reviewer's Responses to Questions

Comments to the Author

5. Review Comments to the Author

Reviewer #1: In the present manuscript Petarli and colleagues aimed to estimate the prevalence of multimorbidity and complex multimorbidity in rural workers and their association with sociodemomographic, occupational, and clinical correlates. The article is quite interesting as it provides estimates based on a population with quite specific lifestyle and occupational exposure to specific agents (e.g. pesticides), which might require targeted interventions/public health policies. What I found quite important to highlight is that in the multivariable model, in contrast to previous literature, almost no socio-demographic correlates (with the exception of age, which is a well-established strong predictor of multimorbidity) remained significant, while all clinical ones were found associated with the likelihood of having multimorbidity and I would suggest authors to elaborate more on this (and possible explanations).

ANSWER: Although there is a large amount of evidence that the occurrence of multimorbidity is higher in females and at low socioeconomic and educational levels [1], the association with socioeconomic variables is very heterogeneous between studies [2]. As with our results, the lack of association with income [3], education and gender [4] has also been documented. Several factors may be related to these results. Among them, we can mention the homogeneity of the rural population investigated in relation to income (92.7% belonged to lower socioeconomic classes - C, D or E), education (89.4% had fewer than 8 years of schooling) and marital status (85.8% were married or living with a partner), compared with the urban population, which generally has more heterogeneous strata, and is therefore more differentiated. This homogeneity of the analyzed population may have compromised the identification of statistically significant differences between strata.

Regarding gender, considering that in rural areas there is limited access to health services [5], there may have been under-reporting in the diagnosis of chronic diseases, especially among women, who generally use health services more often than men. This may have led to a reduction in self-reported disease among women and consequently the absence of statistical association between genders. In addition, the frequency of some diseases is known to vary by gender [6]. In this sense, the methodological differences regarding the type and quantity of diseases to be considered in each study for classification of multimorbidity have a direct influence on the results found by each author [7]. As an example, we can mention the study by Pengpid and Peltzer [3] that identified a higher prevalence of multimorbidity in men due to the inclusion of smoking and alcoholism among the evaluated chronic conditions. The methodology used for disease identification may also have contributed to the difference between the results [7,8]. In the study by Guerra et al. [9], for example, gender was not associated with multimorbidity measured from administrative data, but was associated with self-reported multimorbidity, regardless of the cutoff point adopted. For this reason, the fact that we used both self-reported data as well as biochemical and hemodynamic data may justify the differences in association found when compared to other studies, which mostly use self-reported data.

In addition to methodological differences, the lack of association with some sociodemographic variables may be due to the presence of factors that contribute more closely to the development of multimorbidity, such as waist circumference, previous pesticide poisoning or other factors not intrinsic to agricultural activity within the scope of this study. Further studies involving farmers are needed to better understand the risk factors present in daily agricultural work, thus facilitating the comparison of results. 

All of these considerations were included in the discussion of the article in lines 421 - 456.

Other points:

- Introduction and discussion are sufficiently well-written but to make comparisons even more relevant the cited literature should be enriched with even more recent published papers on the epidmiology of multimorbidity (especially in western countries)

ANSWER: As requested, more recent articles were inserted for comparison purposes. However, it should be noted that a large part of the studies on this subject is conducted with the elderly and our target audience were adult individuals. For this reason, we were careful in choosing the articles to be inserted. It is also worth mentioning that, unfortunately, there are few articles that address multimorbidity in farmers, which made the comparison of results difficult and limited.

- Further details on the sampling strategy should be provided. Authors provide a SS calculation to detect selected outcomes (which should also be expanded with additional information to allow reader to better understand and repeat calculations) but I doubt that this survey was originally based on this SS calculation, please explain and elaborate more on the sampling in general. Additionally, authors do no mention the use of survey weights: does this mean that non-respondent problem was encountered?

ANSWER: For ease of understanding, the sampling session has been rewritten to include more detail on the sample size calculation of the original study and this article.

- The authors adopt an empirical approach for the inclusion of covariates in the multirvariable models which is somehow in contrast with the current tendency to select covariates mostly by literature research. Please justify this approach

ANSWER: The authors chose to insert the variables according to the statistical associations in the chi-square test because they consider this methodology more appropriate given the scarcity of data available in the literature on multimorbidity and its risk factors in a population as specific as farmers. The articles available in the literature, in general, have as target audience very different populations of rural workers, besides presenting methodological differences that may impact the found associations. It is also worth noting that complex multimorbidity has been poorly investigated, and the authors have not identified any studies with this approach in farmers, which limits the theoretical basis on the factors associated with this condition.

Reviewer #2: General comments: Authors have highlighted a very pertinent issues on multimorbidity and its complexities. The overall article content is clear; however,it needs to be reviewed and copy-edited for English language to make language of the article clear, correct before re-submitting.

ANSWER: The article has been reviewed by a specialist in the research area of Proof Reading Service (https://www.proof-reading-service.com/pt/). The certificate provided by the company was attached to the PLOS ONE platform.

Specific comments:

1. Abstract conclusion is not clear and need rephrasing for better clarity.

ANSWER: The Abstract conclusion was rephrased for better clarity.

2. line 100-105: The sentences are confusing and not clear. It would be better to rewrite the sentences with more clarity. It is not clear what authors want to convey by saying "lack of standardization regarding the most effective way to diagnose this condition."

ANSWER: This paragraph was rephrased for better clarity.

3. The study is based on data from "Condition of health and associated factors: a study in farmers of Espírito Santo - AgroSaúdES". As a part of better clarity it would be appropriate to provide a brief description of the parent study: how the sampling was done, how data were collected and what information collected in the parent study.

ANSWER: The methodology has been rewritten to include more detail about the calculation of the original project sample size, how to collect data, and what data was collected as requested by the reviewers. However, to prevent the article from becoming too long, more detail was added especially to the data that was used in this article, since the original project was very large and collected a diverse range of information.

4. Line 120-121: Need rephrasing!

ANSWER: This paragraph was rephrased for better clarity.

5. Sample size calculation: Authors have mentioned that they included 806 sample for their study which would provide 96.9% power to detect the outcome. This is not clear. Authors should provide a detail description of the sample size calculation in a separate section or within the section of study population.

ANSWER: For the analytical developments proposed in this paper the minimum required sample size was recalculated based on an estimated prevalence of multimorbidity in rural populations of 18.6% [10], 3% error, 95% confidence interval, resulting in a minimum required sample of 594 individuals. However, to improve sample representativeness and statistical relevance, we used data from all farmers who participated in the original project. This information was entered in the “sample size calculation” session to facilitate readers' understanding.

6. It would be better to provide a more detail information on how and what information were collected under Anthropometric, hemodynamic and biochemical data.

ANSWER: More details on anthropometric, hemodynamic and biochemical data collection were included in the "data collection" session and in the "variables used in the present study" section. However, to prevent the article from becoming too long, more detail was added to the data that was used in this article, since the original project was very large and collected a diverse range of information.

7. line 157: what is the full form of RSI/WMSD

ANSWER: Full form of RSI/WMSD is Repetitive Strain Injuries/Work Related Musculoskeletal Disorders. This full form was inserted in text.

8. line no.170- The classification of Blood pressure level and not the "pressure level "

ANSWER: Done.

9. Line 197: lifestyle (smoking, physical activity, alcohol consumption)- How these lifestyle factors were elicited and examined?

ANSWER: Further details were added on how lifestyle variables were collected and analyzed in the “Variables selected for this study” session.

10. line 206-208: Authors have categorised waist circumference based on WHO classification and termed it as without cardiovascular risk and increased cardiovascular risk. However, this WHO classification relates to metabolic risk rather than CVD risk.

ANSWER: The term “cardiovascular risk” has been replaced in the text and tables by “metabolic risk” as recommended.

11. Is there any other anthropometric measurements such as height and weight (BMI) were collected in the survey and if collected why authors have not included in the study.

ANSWER: Yes, weight and height (BMI) measurements were collected, but the authors chose to use waist circumference data as they consider that the location of body adiposity is a higher risk factor for disease onset compared to being overweight itself assessed through the IMC [11] In addition, waist circumference is less susceptible to muscle mass interference than BMI and therefore less subject to bias. The authors did not maintain both measures to avoid the effect of collinearity between variables.

12. Line 368: What are the types of diseases when you mentioned the word "disease" in this sentence?

ANSWER: It refers to at least one of the 11 health conditions investigated in the article cited. For the sake of clarity, this passage has been rewritten.

13. References:

a. Line 450: link not working

ANSWER: The link has been changed.

b. References should be checked for uniformity in the formating style and journal names.

ANSWER: References were checked and corrected as suggested by the reviewers.

REFERENCES

1. Violan C, Foguet-Boreu Q, Flores-Mateo G, Salisbury C, Blom J, Freitag M, et al. Prevalence, determinants and patterns of multimorbidity in primary care: a systematic review of observational studies. PLoS One. 2014;9(7):e102149. https://doi.org/10.1371/journal.pone.0102149 PMID: 25048354. 

2. Pathirana TI, Jackson CA. Socioeconomic status and multimorbidity: a systematic review and meta‐analysis. Aust N Z J Public Health. 2018;42(2):186-194. https://doi.org/10.1111/1753-6405.12762 PMID: 29442409. 

3. Pengpid S, Peltzer K. Multimorbidity in Chronic Conditions: Public Primary Care Patients in Four Greater Mekong Countries. Int J Environ Res Public Health. 2017;14(9). pii: E1019. https://doi.org/10.3390/ijerph14091019 PMID: 28878150. 

4. Ba NV, Minh HV, Quang LB, Chuyen NV, Ha BTT, Dai TQ, et al. Prevalence and correlates of multimorbidity among adults in border areas of the Central Highland Region of Vietnam, 2017. J Comorb. 2019; 9: 2235042X19853382. https://doi.org/10.1177/2235042X19853382 PMID: 31192142.

5. Carvalho JN, Cancela MC, Souza DLB. Lifestyle factors and high body mass index are associated with different multimorbidity clusters in the Brazilian population. PLoS One. 2018;20:13(11):e0207649. https://doi.org/10.1371/journal.pone.0207649 PMID: 30458026

6. Zhang R, Lu Y, Shi L, Zhang S, Chang F. Prevalence and patterns of multimorbidity among the elderly in China: a cross-sectional study using national survey data. BMJ Open 2019;9:e024268 https://doi.org/10.1136/bmjopen-2018-024268

7. Abad-Díez JM, Calderón-Larrañaga A, Poncel-Falcó A, Poblador-Plou B, Calderón-Meza JM, Sicras-Mainar A, et al. Age and gender differences in the prevalence and patterns of multimorbidity in the older population. BMC Geriatr. 2014;14:75. https://doi.org/10.1186/1471-2318-14-75 PMID: 24934411. 

8. Aye SKK, Hlaing HH, Htay SS, Cumming R. Multimorbidity and health seeking behaviours among older people in Myanmar: A community survey. PLoS One. 2019;14(7):e0219543. https://doi.org/10.1371/journal.pone.0219543 PMID: 31295287. 

9. Guerra SG, Berbiche D, Vasiliadis HM. Measuring multimorbidity in older adults: comparing different data sources. BMC Geriatr. 2019; 19(1):166. https://doi.org/10.1186/s12877-019-1173-4 PMID: 31200651.

10. Nunes BP, Filho ADPC, Pati S, Teixeira DSC, Flores TR, Camargo-Figuera FA, et al. Contextual and individual inequalities of multimorbidity in Brazilian adults: a cross-sectional national-based study. BMJ Open. 2017;7(6):e015885. https://doi.org/10.1136/bmjopen-2017-015885

11. Uranga RM, Keller, JN. The Complex Interactions Between Obesity, Metabolism and the Brain. Front Neurosci. 2019; 13: 513. https://doi.org/10.3389/fnins.2019.00513 PMID: 31178685.

---

## [Decision Letter · Decision Letter 1]

5 Nov 2019

Multimorbidity and complex multimorbidity in Brazilian rural workers

PONE-D-19-17072R1

Dear Dr. Salaroli,

We are pleased to inform you that your manuscript has been judged scientifically suitable for publication and will be formally accepted for publication once it complies with all outstanding technical requirements.

With kind regards,

Siew Ann Cheong, Ph.D.

Academic Editor

PLOS ONE

Additional Editor Comments (optional):

Reviewers' comments:

Reviewer's Responses to Questions

**Comments to the Author**

1. If the authors have adequately addressed your comments raised in a previous round of review and you feel that this manuscript is now acceptable for publication, you may indicate that here to bypass the “Comments to the Author” section, enter your conflict of interest statement in the “Confidential to Editor” section, and submit your "Accept" recommendation.

Reviewer #1: All comments have been addressed

Reviewer #2: All comments have been addressed

2. Is the manuscript technically sound, and do the data support the conclusions?

Reviewer #1: Yes

Reviewer #2: Yes

3. Has the statistical analysis been performed appropriately and rigorously? 

Reviewer #1: Yes

Reviewer #2: Yes

4. Have the authors made all data underlying the findings in their manuscript fully available?

Reviewer #1: Yes

Reviewer #2: Yes

5. Is the manuscript presented in an intelligible fashion and written in standard English?

Reviewer #1: Yes

Reviewer #2: Yes

6. Review Comments to the Author

Reviewer #1: Authors have addressed all my comments. A very minor comment is that p value = 0.000 should be replaced with p value<0.0001

Reviewer #2: (No Response)

7. PLOS authors have the option to publish the peer review history of their article (what does this mean?). If published, this will include your full peer review and any attached files.

Reviewer #1: Yes: Raffaele Palladino

Reviewer #2: No

---

## [Editor Report · Acceptance letter]

7 Nov 2019

PONE-D-19-17072R1 

Multimorbidity and complex multimorbidity in Brazilian rural workers 

Dear Dr. Salaroli:

I am pleased to inform you that your manuscript has been deemed suitable for publication in PLOS ONE. Congratulations! Your manuscript is now with our production department. 

With kind regards,

on behalf of

Dr. Siew Ann Cheong 

Academic Editor

PLOS ONE